# Increased IGFBP2 Levels by Placenta-Derived Mesenchymal Stem Cells Enhance Glucose Metabolism in a TAA-Injured Rat Model via AMPK Signaling Pathway

**DOI:** 10.3390/ijms242216531

**Published:** 2023-11-20

**Authors:** Dae-Hyun Lee, Hyeri Park, Jun-Hyeong You, Jin Seok, Dong-Wook Kwon, Young-Ran Kim, Gi-Jin Kim

**Affiliations:** 1Department of Bioinspired Science, CHA University, Seongnam-si 13488, Republic of Korea; ldh92426@gmail.com (D.-H.L.); hyeyeyeri@gmail.com (H.P.); yjh950210@gmail.com (J.-H.Y.); k9354680@gmail.com (D.-W.K.); 2PLABiologics Co., Ltd., Seongnam-si 13522, Republic of Korea; 3Department of Obstetrics and Gynecology, University of Chicago, 5841A. Maryland Ave., Chicago, IL 60637, USA; 4Department of Obstetics and Gynecology, CHA Bundang Medical Center, Seongnam-si 13496, Republic of Korea; happyimam@naver.com

**Keywords:** IGFBP2, AMPK, ovarian dysfunction, mitochondria, placenta-derived mesenchymal stem cells

## Abstract

The insulin resistance caused by impaired glucose metabolism induces ovarian dysfunction due to the central importance of glucose as a source of energy. However, the research on glucose metabolism in the ovaries is still lacking. The objectives of this study were to analyze the effect of PD-MSCs on glucose metabolism through IGFBP2–AMPK signaling and to investigate the correlation between glucose metabolism and ovarian function. Thioacetamide (TAA) was used to construct a rat injury model. PD-MSCs were transplanted into the tail vein (2 × 10^6^) 8 weeks after the experiment started. The expression of the IGFBP2 gene and glucose metabolism factors (e.g., AMPK, GLUT4) was significantly increased in the PD-MSC group compared to the nontransplantation (NTx) group (* *p* < 0.05). The levels of follicular development markers and the sex hormones AMH, FSH, and E2 were also higher than those in the TAA group. Using ex vivo cocultivation, the mRNA and protein expression of IGFBP2, AMPK, and GLUT4 were significantly increased in the cocultivation with the PD-MSCs group and the recombinant protein-treated group (* *p* < 0.05). These findings suggest that the increased IGFBP2 levels by PD-MSCs play an important role in glucose metabolism and ovarian function through the IGFBP2–AMPK signaling pathway.

## 1. Introduction

One of the most common endocrine disorders, polycystic ovarian syndrome (PCOS), has a global incidence range of 6–20% and is associated with women’s anovulation and hyperandrogenemia, as well as long-term health issues [1]. It is characterized by hyperandrogenism, metabolic syndrome, insulin resistance (IR), infertility, anovulation, and ovarian cysts [2]. For this reason, PCOS is now considered a metabolic disorder, and women with PCOS have an increased risk of gestational diabetes, impaired glucose tolerance, and type 2 diabetes [3].

Glucose homeostasis is tightly regulated by several sugar transporters that exhibit different substrate specificities. Therefore, abnormal glucose metabolism affects cellular function and survival. Most mammalian cells use glucose predominantly for adenosine triphosphate (ATP) synthesis via the tricarboxylic acid (TCA) pathway, followed by oxidative phosphorylation. Additionally, the energy produced is essentially used for oocyte development [4]. In the ovaries, the follicles can be divided into primordial follicles, primary follicles, secondary follicles, and mature follicles according to the morphologic phenotype being in different developmental stages [5]. Therefore, the process of fertilization requires a great deal of energy because many factors change the morphology and function of ovarian follicles via glucose metabolism [6,7].

Mitochondria affects all aspects of mammalian reproduction by converting the chemical energy of organic matter into ATP via intracellular respiration [6]. They are essential for optimal maturation, fertilization, and embryonic development. Therefore, mitochondrial dysfunction causes a decrease in oocyte development. Additionally, mitochondrial deoxyribonucleic acid (mtDNA) copy numbers at an appropriate level are crucial for maintaining mitochondrial function; that is, changes in the mtDNA copy number reflect mitochondrial disorders caused by environmental oxidants [7]. Changes in the levels of ATP in the follicular fluid of patients with polycystic ovarian syndrome (PCOS) are accompanied by disordered redox potential and enhanced reactive oxygen species (ROS) levels in cumulus cells (CCs) [8]. Abnormal ovarian mitochondrial function caused by lipotoxicity is directly related to reproductive dysfunction. An increase in the lipid content in the ovaries has been observed, and this can lead to lipotoxicity (e.g., ER stress, mitochondrial dysfunction, and apoptosis) and reproductive dysfunction because an appropriate quality and quantity of mitochondria in the oocytes are needed for proper fertilization and development [9].

Among several factors present in ovarian mitochondria, AMP-activated protein kinase (AMPK), which is a serine–threonine heterotrimeric kinase, is activated when energy fails to produce more energy to maintain ATP homeostasis [10]. Moreover, AMPK controls the crosstalk between energy balance and reproduction, resulting in downregulated glucose metabolism, which can trigger defects in ovulation and lower oocyte quality [11]. Additionally, AMPK signaling is activated by IGFBP (insulin-like growth factor binding protein) family ligands complexed with IGFs as regulators of IGF actions in metabolism and growth [12]. The function of IGFBPs is related to metabolic homeostasis, insulin resistance, diabetes, and obesity. In particular, IGFBP2 is one of the key factors in the processes of follicle growth, differentiation, and maturation in an IGF-dependent and IGF-independent manner [13]. Kamangar et al. suggest that the upregulation of IGFBP2 controlled by estradiol in follicles or extrafollicular tissues and is closely linked to the ovarian follicular pathophysiology via the regulation of energy [14].

Hormone replacement therapy (HRT), metformin treatment, and immunomodulation therapy are currently used to treat ovarian dysfunctions. However, this therapy is associated with the risks of temporal effects, breast cancer, and cardiovascular disease [15]. For example, metformin therapy has become widely used for the correction of metabolic patients to restore their organ function. It is most effective in treating PCOS patients with metabolic disorders such as T2DM, dyslipidemia, and severe IR. However, gastrointestinal side effects of metformin have been reported in several cases, and it has a risk of cancer.

In this study, we suggest stem cell therapy as a new therapeutic strategy for metabolic disorders such as PCOS. Mesenchymal stem cells (MSC) can serve as a new biomaterials for the treatment of human reproductive dysfunction, including female infertility. Additionally, the clinical application of MSCs holds great promise for the treatment of infertility or ovarian insufficiency and for improving reproductive health for a significant number of women worldwide [16]. Recently, placenta-derived mesenchymal stem cells (PD-MSCs) have been shown to have several advantages because they have a high rate of self-renewal activity, strong immunosuppressive and multilineage differentiation potential, and no ethical problems [17]. In our previous reports, PD-MSCs enhanced the ovarian function of injured ovarian tissues of an ovariectomized rat model through increased antioxidant effects by activating phosphoinositide 3 kinase–forkhead box O (PI3K-FOXO) signaling and vascular remodeling via vascular endothelial growth factor (VEGF) signaling [18,19]. However, the regulatory mechanisms of PD-MSCs on glucose metabolism are still unclear in ovarian physiology. Therefore, the objectives of the present study were to demonstrate the effect of PD-MSCs on the ovarian function of a TAA-injured rat model with metabolic disorder and to investigate the effect of PD-MSCs on glucose metabolism through the AMPK signaling pathway in TAA-injured rat ovaries. If PD-MSCs trigger glucose homeostasis in a TAA-injured rat model with metabolic disorder, they could regulate the expression of IGFBP2 in injured ovarian tissues. Finally, we evaluated whether IGFBP2 cytokine production by PD-MSCs could enhance ovarian function via the AMPK signaling pathway in glucose metabolism.

## 2. Results

### 2.1. Thioacetamide Induces Ovarian Dysfunction and Insulin Resistance and Impairs Glucose Metabolism

To evaluate TAA-induced insulin resistance and impaired glucose metabolism, we performed a homeostatic model assessment for insulin resistance (HOMA-IR) based on the glucose and insulin levels in thioacetamide-injured rat serum. In the nontransplantation group (NTx), the glucose levels increased more than those in the normal group (Nor; * *p* < 0.05), whereas decreased in the PD-MSC transplantation group (Tx, Figure 1A; * *p* < 0.05). In the case of the insulin levels, their concentration patterns were similar to those of glucose (Figure 1B; * *p* < 0.05). Generally, TAA is a thiosulfur compound and is widely known as a hepatotoxin that triggers liver damage by generating ROS [20].

To confirm, we measured the ALB, total bilirubin, ALT, and AST levels in thioacetamide injury blood serum to check the liver function in the TAA-injured rat model. The albumin level decreased in the NTx group compared with the normal group (* *p* < 0.05). In contrast, the total bilirubin, ALT, and AST levels increased in the NTx group and decreased in the Tx group (Figure 1C–F; * *p* < 0.05). The homeostasis model assessment (HOMA) for IR is the most widely used surrogate marker in clinical and epidemiologic studies, which can reduce the time and cost and is relatively accurate [21].

To evaluate the insulin resistance state, we performed an HOMA-IR and homeostatic model assessment for insulin sensitivity (HOMA-IS). The HOMA-IR value in the NTx group was 28.1777, while that in the Tx group was 7.85098 (Figure 1G; * *p* < 0.05). Thus, TAA induced metabolic imbalance in this model, although the PD-MSCs ameliorated insulin resistance. Additionally, we achieved cross-validation using the HOMA-IS index (Figure 1H). We analyzed the morphologies of liver and ovarian tissues. In the liver, we observed liver fibrosis. Additionally, in the ovarian NTx tissue, the ratios of the ovary weight to body weight were all decreased compared to those in the Nor group (Figure 1I,J; * *p* < 0.05). Both the NTx and Tx groups had ovary/body weight ratios that were similar to those of the Nor group (Figure 1K). These data were similar to previous reports showing that PD-MSC transplantation led to histological recovery of the liver and ovaries in TAA-injured rats [22].

Additionally, we investigated cell apoptosis and autophagy in TAA-induced rat ovarian tissues. The TUNEL score indicating cell apoptosis increased in the NTx group compared with the Nor group (* *p* < 0.05); however, the score decreased after PD-MSC Tx (Figure 1L,M). Additionally, we confirmed the autophagosome formation using immunofluorescence staining (Figure 1N,O) and Western blotting (Figure 1P–S). In Figure 1, the white arrow indicates type II programmed cell death and autophagic markers, specifically LC3 type II (LC3B, active form). Additionally, it is shown that the LC3B protein expression decreased in the NTx group but increased in the Tx group (Figure 1N,O; * *p* < 0.05). Moreover, we confirmed that the LC3B and BECN1 protein expression decreased in the NTx group compared with the normal group, while their expression was increased in the Tx group (Figure 1P–R; * *p* < 0.05). However, mTOR expression, which is a negative feedback factor for autophagy, was increased in the NTx group compared with the normal group and decreased in the Tx group (Figure 1S; * *p* < 0.05). These findings suggest that TAA induced metabolic imbalance and cell death in ovarian tissues, although PD-MSC transplantation ameliorated metabolic imbalance and inhibited cell death in TAA-injured ovaries.

### 2.2. Glucose Metabolism and Insulin Pathway Signaling Activated by PD-MSCs through AMPK Signaling Pathway

In the glucose metabolism pathway, insulin-like growth factor-1 (IGF-1) activates AMPK signaling and insulin-like growth factor binding protein-2 (IGFBP2) regulates glucose metabolism [12]. Therefore, we hypothesized that the increased level of IGFBP2 induced by PD-MSCs would activate glucose metabolism.

To demonstrate activating glucose metabolism through IGFBP2, we analyzed the expression of GLUT4 gene expression in sectioned TAA-injured rat ovary tissue. The expression of GLUT4 decreased in the ovarian tissues of the NTx group but increased in the Tx group. Interestingly, their expression was significantly increased in the granulosa layers after PD-MSC transplantation compared to that in the NTx group (Figure 2A,B; * *p* < 0.05). Additionally, the expression pattern of IGFBP2 was similar to the GLUT4 expression patterns in each group (Figure 2C,D; * *p* < 0.05).

To confirm that PD-MSCs affect insulin pathway signaling, the expression levels of insulin receptor substrate 1 (IRS1) and insulin receptor (IR) were analyzed by using qRT-PCR. The expression of these genes was significantly decreased in the NTx group (* *p* < 0.05) but increased in the Tx group (Figure 2E,F; * *p* < 0.05).

Froment et al. suggested that evaluating the activation of AMPK could be considered a marker for infertility cases related to hormonal imbalances and metabolic disorders such as PCOS [11]. We confirmed that the phosphorylated AMPK was decreased in the NTx group compared to the normal and Tx groups (Figure 2G; * *p* < 0.05). In addition, the expression of metabolic markers, including sirtuin 1 (SIRT1), GLUT4, IGFBP2, p-FOXO3a, p-PI3K, p-AKT (serine/threonine kinase 1), and p-IR, was increased in the Tx group compared to the NTx group (Figure 2G–O; * *p* < 0.05).

Finally, the ATP production and mitochondrial DNA copy number (mtDNA) were also increased in the ovarian tissues after PD-MSC transplantation compared to the NTx group (Figure 2P,Q; * *p* < 0.05). These results indicated that increasing IGFBP2 expression via PD-MSC transplantation triggers the activation of glucose metabolism and the insulin pathway, resulting in increases in mitochondrial ATP production and mitochondrial copy number in ovarian tissues to restore the function of injured ovaries.

### 2.3. Effect of PD-MSCs on Ovarian Function in the Ovaries of the TAA-Injured Rats

Abnormalities in follicle development affect not only the endocrine environment but also the quality and number of follicles [23,24]. Notably, glucose regulation by the AMPK signaling pathway improves follicular development, and the AMPK-SIRT1 molecular pathway was a major factor in the regulation of insulin resistance in a PCOS mouse model [25].

To investigate ovarian function, the anti-Mullerian hormone (AMH), estradiol (E2), follicle-stimulating hormone (FSH), and testosterone hormone levels were analyzed using an enzyme-linked immunosorbent assay (ELISA) in a TAA-injured rat model with or without PD-MSC transplantation. The AMH, E2, and FSH hormone levels were increased in the Tx group compared to the NTx group with the endocrine environment disrupted by TAA administration; on the other hand, testosterone was decreased in the Tx group compared to the NTx group (Figure 3A–D; * *p* < 0.05). These data indicate that PD-MSC transplantation enhances the expression of markers for ovarian reserve, such as AMH, E2, and FSH, and decreases the high levels of testosterone.

To confirm the effect on follicular development, we evaluated serial ovarian tissues from each group using H&E staining (Figure 3). Then, each follicle stage was counted by three different people. As shown in the data, the TAA treatment increased the number of atretic follicles and decreased the total number of follicles. However, PD-MSC Tx partially protected follicles damaged by TAA, especially the primordial follicles (Figure 3F). The expression of follicular development markers, including Nanos homolog 3 (Nanos3), NOBOX oogenesis homeobox (NOBOX), bone morphogenetic protein 15 (BMP15), and epidermal growth factor receptor (EGFR), was increased in the Tx group compared to the NTx group (Figure 3G–K; * *p* < 0.05). These results suggest that PD-MSC transplantation alleviates ovarian dysfunction by increasing markers of follicular development.

### 2.4. The Increase in IGFBP2 Secreted by PD-MSCs Triggers Glucose Metabolism in a TAA-Injured Rat Ovary Model

Generally, IGFBP2 is decreased in metabolic disorders, in which several risk factors lead to decreased IGFBP2 expression levels [26]. To confirm that IGFBP2 secreted by PD-MSCs triggers the glucose metabolism signaling pathway, we investigated the paracrine effect of PD-MSCs and the combination of IGF-1 and IGFBP2. Based on the MTT assay results, the optimal concentrations of PPP (100 ng/mL), which is an IGF-1 inhibitor, and recombinant IGFBP2 (100 ng/mL) were determined. The ovaries injured by TAA (70 mM) were cocultured with PD-MSCs, PPP, or IGFBP2 for 24 h (Figure 4A). Then, we examined the expression of IGFBP2 cytokine-mediated glucose metabolism markers with or without PD-MSC cocultivation, an inhibitor, and a recombinant.

The expression of glucose-metabolism-related markers (e.g., IGFBP2, AMPK, SIRT1, and GLUT4) and insulin signaling pathway intermediaries (e.g., IGF-1 and IR) was decreased in ovarian tissues injured by TAA but increased with PD-MSC cocultivation and recombinant IGFBP2 treatments (Figure 4B–G). However, their expression levels in the PD-MSC cocultivation group were higher than in the recombinant IGFBP2 treatment group.

Finally, their expression in ovarian tissues exposed to PPP treatment was dramatically decreased, although PD-MSC cocultivation restored their decreased expression patterns. The protein expression of AMPK, GLUT4, and IGFBP2 was increased in the group cocultured with recombinant PD-MSCs but decreased in the group cocultured with PPP (IGF-1 inhibitor) (Figure 4H–J). This result indicated that the IGFBP2/IGF-1-mediated signaling pathway activated by PD-MSCs could trigger AMPK signaling and the insulin pathway to activate glucose metabolism. The data are presented as the mean ± S.D. Statistical significance was determined by using a one-way ANOVA (* *p* < 0.05, Nor vs. TAA treatment; # *p* < 0.05, NTx vs. cocultivation with PD-MSCs, recombinant IGFBP2, or inhibitor; $ *p* < 0.05 cocultivation with PD-MSCs vs. recombinant IGFBP2; ## *p* < 0.05 inhibitor vs. inhibitor + PD-MSCs).

### 2.5. The Activation of Glucose Metabolism Markers Was Enhanced by the AMPK Signaling Pathway in Granulosa Cells (In Vitro)

Referring to Tao et al., unbalanced AMPK and SIRT1 activation induces abnormal energy metabolism disorders [25]. To investigate whether glucose metabolism activation by the AMPK signaling pathway could affect theca cells and granulosa cells, we used a cocultivation system using the KGN, which comprises human granulosa-like cells [27] and theca cell lines (Figure 5A). Then, we analyzed the mRNA and protein levels of glucose metabolism markers in granulosa cells and theca cells. In theca cells, the mRNA expression levels of IGFBP2, IGF-1, AMPK, and SIRT1 were decreased in the TAA treatment group, although their expression was increased in the PD-MSC cocultivation group (Figure 5B–E; * *p* < 0.05). Furthermore, their expression was like that of the granulosa cell system (Figure 5F–J; * *p* < 0.05). Additionally, p-AMPK and GLUT4 expression was significantly increased in the PD-MSC cocultivation group compared to the TAA treatment group (Figure 5K–M; * *p* < 0.05). Therefore, these results indicated that PD-MSC cocultivation activates glucose metabolism through activation of the IGFBP2-mediated AMPK pathway in granulosa cells and theca cells.

## 3. Discussion

The production of ATP, an energy source generated through the glucose metabolic pathway, is essential for various dynamic cellular events, including energy production, hormone synthesis, and homeostasis in the ovaries [28]. In addition, activated glucose metabolism in the ovary can affect follicular development and maturation, and impaired glucose metabolic regulation could lead to ovarian dysfunction, including the inhibition of follicular development [29].

To address this issue, numerous investigations have revealed a direct connection between AMPK and IR, the inflammatory response, and glycolipid metabolism, all of which have a positive impact on glucose homeostasis, insulin tolerance, and reproduction [10,11]. In previous reports, we demonstrated that PD-MSC transplantation restored ovarian function in an ovariectomized rat model through increased antioxidant effects in the mitochondria of injured ovarian tissues [30]. It has been demonstrated that PD-MSCs have an impact on injured ovary tissue via antioxidant effects.

Furthermore, we extended the therapeutic effect of PD-MSCs on metabolic disorder via IGFBP2 cytokines secreted by PD-MSCs. Based on the above results, we formulated a hypothesis that glucose activates primordial follicles through the IGFBP2–AMPK signaling pathway. Due to the critical importance of maintaining a balance between glucose and insulin homeostasis for cell survival and function, an imbalance in chorionic glucose homeostasis can lead to various diseases, including liver disease and polycystic ovary syndrome (PCOS) [31]. According to our preliminary studies, hepatic tissue damage in rats injured with TAA not only leads to a structural increase in collagen accumulation but also causes functional issues with glucose storage due to abnormal glucose control in the liver tissue [20]. Furthermore, another recent study has suggested that optimal stem cell treatment enhances both the structure and function of the liver [32]. These pathological observations suggest that the restoration to normal conditions, facilitated by the antifibrotic effects and maintenance of glucose homeostasis, could potentially be attained through the transplantation of PD-MSCs.

In addition, in our earlier reports, we demonstrated that PD-MSCs can ameliorate metabolic imbalance not only in TAA-injured rat livers but also in ovaries [22]. Nevertheless, in the absence of a mechanistic study on how and by which factor blood glucose is regulated by stem cell transplantation, this study represents the first report to demonstrate that IGFBP2 secreted from PD-MSCs activates the AMPK signaling system. For the study reported here, we induced ovarian dysfunction using thioacetamide (TAA). To confirm the injured ovary model, we analyzed biochemical characteristics using rat blood. Remarkably, the data in Figure 1 demonstrate that the transplantation of PD-MSCs enhances the metabolic imbalance status. Additionally, we observed an antiapoptotic effect and an augmentation of autophagy in ovarian tissue following the transplantation of PD-MSCs. This observation implies that TAA leads to both metabolic dysfunction and ovarian weight loss. However, the transplantation of PD-MSCs seems to reverse this imbalance. Moreover, this finding suggests the potential for enhancing the autophagy mechanism through PD-MSCs.

To demonstrate that IGFBP2 expression and glucose metabolism increased through the transplantation of PD-MSCs, we conducted immunostaining, qRT-PCR, and Western blot measurements. As shown in Figure 2, we observed increases in phosphorylated AMPK and glucose transporter type 4 (GLUT4) expression in the PD-MSC transplantation (Tx) group compared to the nontransplantation (NTx) group. Additionally, IGFBP2 gene expression was found to be increased in the Tx group. These results suggest that PD-MSC transplantation increased markers related to glucose and the insulin signaling pathway. Based on the above results, we speculate that IGFBP2 cytokines regulate this signaling. Additionally, we infer that this biological phenomenon occurs within granulosa cells, as evidenced by the stronger expression of GLUT4 in the granulosa cell layer compared to the theca cell layer. 

According to the Rotterdam criteria and our colleagues, the primary diagnostic criteria for PCOS include clinical or biochemical hyperandrogenism, oligomenorrhea, or amenorrhea associated with chronic anovulation, morphological features indicative of PCOS, and the manifestation of insulin resistance (IR) [2,33,34]. PCOS is a relatively common but poorly understood disorder with both reproductive and metabolic components. Furthermore, the mechanism by which insulin resistance diminishes endocrine and reproductive functions in ovarian dysfunction remains unclear. Hence, the improvement of insulin resistance through stem cell therapy may prevent metabolic complications in women with PCOS. 

Several studies have demonstrated that mesenchymal stem cell therapy alleviates metabolic dysfunction and restores fertility in mouse models of PCOS. [35,36]. As demonstrated in Figure 3, our study affirms that the transplantation of PD-MSCs leads to an increase in markers associated with follicular development, including AMH, FSH, and estrogen. Additionally, elevated testosterone levels were confirmed in TAA-injured rats compared to normal rats, reflecting one of the phenotypes of PCOS, and testosterone levels were ameliorated with PD-MSC transplantation. Interestingly, following PD-MSC transplantation, we observed that both primordial and antral follicles were protected, while the number of atretic follicles decreased.

This suggests that PD-MSC transplantation could prevent the development of antral follicles into atretic follicles. Taken together, these results indicate that PD-MSCs transplantation improves follicular development and protects each follicle.

In a previous report, we reported that PD-MSCs secreted the IGFBP2 cytokine based on a cytokine array [37]. Building on our earlier report, we showed that IGFBP2 activates glucose metabolism and insulin pathway signaling through the IGFBP2–AMPK signaling pathway, mediated by a paracrine effect. To confirm that the IGFBP2 cytokine increases glucose metabolism, we conducted ex vivo experiments. As shown in Figure 4, IGFBP2 and glucose metabolism markers were suppressed with the inhibitor treatment but activated with the treatment of recombinant IGFBP2. This result indicated that the IGFBP2 cytokine activated glucose metabolism and insulin pathway signaling. Finally, the enhancement of glucose metabolism through the IGFBP2–AMPK signaling pathway improved the mitochondrial function in oocytes.

Additionally, granulosa cells, known for their high glycolytic activity level, generate pyruvate for metabolism via oocyte mitochondria [38]. To confirm the glucose metabolism activation in granulosa cells, we conducted an in vitro experiment. As shown in Figure 5, the expression of markers for glucose metabolism in both theca and granulosa cell was activated by the PD-MSC cocultivation. Collectively, we conclude that the glucose metabolism was more activated in the granulosa cell layer than the theca cell layer. This observation underscores the significance of the granulosa cell layer as the primary contributor to ATP generation by oocyte mitochondria [39]. 

Additional investigations are warranted to elucidate the intricacies of glucose metabolism activation between granulosa cells and oocytes. For future clinical experiments, employing larger sample sizes is imperative to enhance the statistical robustness.

## 4. Materials and Methods

### 4.1. Cell Culture

Placentas obtained at term (37 gestational weeks) in women who did not experience any obstetric, perinatal, or surgical problems were used. The IRB at CHA General Hospital, Seoul, Korea, approved the sample collection and usage for the study (IRB 07–18). Written informed consent was obtained from all women who participated. The PD-MSCs were isolated using a previously described method [40]. Alpha-modified minimal essential medium (α-MEM; HyClone, Logan, UT, USA) was used to maintain the PD-MSCs. This medium was supplemented with 10% fetal bovine serum (FBS; Gibco, Carlsbad, CA, USA), 1% penicillin/streptomycin (P/S; Gibco, Carlsbad, CA, USA), and 25 ng/mL of human fibroblast growth factor-4 (hFGF-4; PeproTech, Rocky Hill, NJ, USA). The cells were maintained at 37 °C in a humid atmosphere containing 5% CO^2^ for 24 h.

### 4.2. The Human GC-like Tumor Cell Line KGN

Riken Bio (Saitama, Japan) provided the human granulosa-like tumor cell line KGN. At 37 °C in a humidified environment with 5% CO_2_, the KGN cells were cultivated in DMEM/F12 media (Gibco, Carlsbad, CA, USA) supplemented with 10% fetal bovine serum (Gibco, Carlsbad, CA, USA).

### 4.3. Primary Cell Culture

Female Sprague–Dawley rats aged seven weeks were aseptically removed from their ovaries and sacrificed via CO_2_ inhalation. The isolated ovaries were placed in McCoy’s 5A medium (Gibco, Carlsbad, CA, USA) containing 2 mM L-glutamine (Sigma, St. Louis, MO, USA) and 1% penicillin/streptomycin (P/S; Carlsbad, CA, USA). After the ovaries were cleared of any fat that stuck to them, the granulosa and red blood cells were released by puncturing them with a 26 1/2-gauge needle in a glass Petri plate. A 100-μm cell strainer was used to strain the residual ovarian tissue using medium. The ovarian tissue on the cell strainer was vortexed, and then it was treated for 1 h at 37 °C in medium containing 4 mg/mL of collagenase (Sigma, St. Louis, MO, USA), 40 mg/mL of deoxyribonuclease I (Sigma, St. Louis, MO, USA), and 10 mg/mL of BSA (RDT). After being liberated by the digestion process, the cells were strained through a 40-μm cell strainer, centrifuged for 5 min at 1000 rpm, and then it was washed three times in the McCoy’s 5A medium to remove any leftover enzymes. Ultimately, the cells were kept in a humidified environment with 5% CO_2_ at 37 °C.

### 4.4. Ex Vivo Culture System

BD Biosciences (Franklin, NJ, USA) provided the 24-well insert system that was used to create the ex vivo cocultivation system. Every ovary from the rats was removed, and PD-MSCs were cocultured with them. The ovaries were placed on a 24-well plate and covered with Matrigel (Corning, NY, USA). Following their removal from the ovaries, the extracellular fat reserves were washed with Dulbecco’s phosphate-buffered saline (DPBS, Welgene, Gyeongsan-Si, Republic of Korea) and α-MEM (α-MEM; HyClone, Logan, UT, USA), which included 1% penicillin/streptomycin and 10% FBS (Gibco, Carlsbad, CA, USA). Every one of the extracted and chopped ovaries was inserted in a 24-well plate, and 1 × 10^4^ PD-MSCs were seeded on an insert that was put on top of the 24-well plate with the medium and ovary. Next, the cocultivated ovaries were gathered for examination. The ovaries were positioned on Matrigel in 24-well plates and treated with recombinant human IGFBP2 (150 ng/mL, PeproTech) and PPP (IGF-1 inhibitor, 150 ng/mL, Sigma, St. Louis, MO, USA) in the culture media in order to examine the effects of IGFBP2 administration, IGF-1 inhibition, and PD-MSC cocultivation. The samples were gathered for examination after 24 h.

### 4.5. TAA Animal Modeling and PD-MSC Transplantation

All animal studies at the CHA laboratory animal research facility in Korea were approved by the Institutional Animal Care and Use Committee (IACUC 220044). Eight-week-old female Sprague–Dawley rats (Orient Bio Inc., Seongnam, Republic of Korea) were housed in a pathogen-free, air-conditioned facility at room temperature (21 °C) with a 12 h light-dark cycle. Thioacetamide (TAA, 300 mg/kg, Sigma-Aldrich, St. Louis, MO, USA) was injected intraperitoneally (i.p.) twice a week for 12 weeks, causing ovarian damage. At eight weeks, naïve cells (2x10^6^) were injected intravenously. After twelve weeks, the rats were killed.

### 4.6. Blood Chemistry Test

The Southeast Medi-Chem Institute (Busan, Republic of Korea) measured the serum levels of insulin, glucose, albumin (ALB), total bilirubin, aspartate aminotransferase (AST), and alanine aminotransferase (ALT).

### 4.7. RNA Isolation and Quantitative Real-Time Polymerase Chain Reaction

Ovarian tissues from rats were extracted and rapidly frozen. The total RNA was isolated after the tissues had been homogenized. Each sample group received 0.2 mL of chloroform and 1 mL of TRIzol reagent (Invitrogen, Carlsbad, CA, USA) before centrifugation to separate the supernatant. The particles were obtained by washing the separated supernatant with isopropyl alcohol or ethyl alcohol and discarding it. The pellet was subsequently dissolved in water treated with DEPC (Invitrogen) at 60 °C. The concentration of total RNA was determined using a Nanodrop spectrophotometer (Thermo Fisher Scientific, Waltham, Maryland, USA). Superscript III reverse transcriptase (Invitrogen) was used to convert whole RNA into cDNA. The following are the PCR requirements for cDNA synthesis: 5 min at 65 °C, 1 min at 4 °C, 60 min at 50 °C, and 15 min at 72 °C. The qRT-PCR was performed using FS Universal SYBR Green Master ROX and cDNA (Roche, Basel, Switzerland). Subsequently, the cDNA was amplified using PCR under the following conditions: 5 s at 95 °C, followed by 40 cycles of 5 s at 95 °C and 30 s at 60 °C. The sequences of the qRT–PCR primers are listed in Table 1. Each sample was examined in triplicate, with rat GAPDH used as the internal control for standardization.

### 4.8. Protein Isolation and Western Blotting

Rat ovarian tissues from each group were homogenized and lysed on ice using RIPA buffer (Sigma-Aldrich, USA), which contained a phosphatase inhibitor cocktail and a protease inhibitor cocktail (genDEPOT, Barker, TX, USA). Using a bicinchoninic acid test (BCA) protein assay kit (Thermo Fisher Scientific, Waltham, MD, USA), the protein concentrations from individual rats were measured and standardized. Protein extracts of equal concentration were separated via electrophoresis on 9% sodium dodecyl sulfate–polyacrylamide gels (SDS-PAGE). Using a turbo-transfer system, the separated proteins were transferred onto a polyvinylidene difluoride (PVDF) membrane (Bio-Rad Laboratories, Hercules, CA, USA). The membranes were blocked in blocking solution (5% BSA) for 1 h at room temperature. The membranes were then exposed to the primary antibody in 2% BSA overnight at 4 °C. The following antibodies were mixed with 2% BSA and incubated at 4 °C: rabbit anti-AMPK (phospho 2535S, Cell Signaling Technology, Danvers, MA, USA) diluted 1:500, rabbit anti-AMPK (total 2532S, Cell Signaling Technology) diluted 1:1000, mouse anti-GLUT4 (ab216661, abcam, Boston, Massachusetts, USA) diluted 1:1000, rabbit anti-total mTOR (7C10; 2983S, Cell Signaling Technology) diluted 1:1000, rabbit anti-mTOR (phospho S2448; ab109268, abcam) diluted 1:1000, rabbit anti-LC3B (2775S, Cell Signaling Technology) diluted 1:1000, rabbit anti-BECN1 (SC11427, Santacruz, Dallas, Texas, USA) diluted 1:1000, rabbit anti-SIRT1 (BS-0921R, Bioss, Woburn, Massachusetts, USA) diluted 1:1000, rabbit anti-FOXO3a (phospho 9466S, Cell Signaling Technology) diluted 1:500, rabbit anti-FOXO3a (12829S, Cell Signaling Technology) diluted 1:1000, rabbit anti-PI3K (3811S, Cell Signaling Technology) diluted 1:2000, rabbit anti-AKT (phospho 927S, Cell Signaling Technology) diluted 1:1000, rabbit anti-AKT (9272S, Cell Signaling Technology) diluted 1:1000, rabbit anti-IGFBP2 (BS-1108R, Bioss) diluted 1:2000, rabbit anti-INSR (PA5-27334, Invitrogen) diluted 1:2000, mouse anti-insulin receptor (phospho SC-81500, Santa Cruz) diluted 1:500, mouse anti-Nobox (SC-390016, Santa Cruz) diluted 1:1000, rabbit anti-BMP15 (MBS2516631, Mybiosource, San Diego, CA, USA) diluted 1:500, rabbit anti-EGF receptor (2232S, Cell Signaling Technology) diluted 1:1000, rabbit anti-Nanos3 (ab70001, abcam) diluted 1:1000, and rabbit anti-GAPDH (LF-PA0018, Abfrontier, Seoul, Republic of Korea) diluted 1:2000. After incubation, the membranes were washed with 1× Tris-buffered saline–Tween 20 (TBS-T) and incubated with a secondary antibody (1:10,000) at room temperature for 1 h according to the manufacturer’s instructions. The membranes were washed before exposure to the Clarity Western ECL kit (Bio-Rad Laboratories, Hercules, CA, USA) for 5 min at room temperature. The Chemi Doc XRS+ imaging technology was used to identify the protein bands (Bio-Rad Laboratories). ImageJ software (Wayne Rasband, Bethesda, Maryland, USA) was used to analyze the bands. The fold change value of intensity is the comparative value of gene expression.

### 4.9. The gDNA Extraction and Mitochondrial DNA Copy Number Analysis

Genomic DNA (gDNA) was isolated from the ovarian tissue using the QIAamp DNA Mini Kit (Qiagen, Valencia, CA, USA). Next, 700 µL of the lysis solution containing 20 g/mL protease K (Cosmo Gene Tech., Seoul, Korea) was added to the homogenized tissues. Then, the samples spent a single hour at 60 °C in an incubator. The gDNA was extracted in accordance with the manufacturer’s instructions after incubation. To determine the concentration of gDNA, a Thermo Scientific Nanodrop spectrophotometer was utilized. The mitochondrial DNA (mtDNA) copy number and gene expression were investigated via qRT-PCR using the 2× TaqMan universal master mix (Applied Biosystems, CA, USA). The internal control for the normalization of the qRT-PCR was nuclear DNA. The following primers were used: rat mitochondrial D-loop, F: 5′-GGT TCT TAC TTC AGG GCC ATC A-3′, R: 5′-GAT TAG ACC CGT TAC CAT CGA GAT-3′, probe, JOE-TTG GTT CAT CGT CCA TAC GTT CCC CTT A-3′; rat β-actin, F: 5′-GGG ATG TTT GCT CCA ACC AA-3′, R: 5′-GCG CTT TTG ACT CAA GGA TTT AA-3′, probe, FAM 5′-CGG TCG CCT TCA CCG TTC CAG TT-3′.

### 4.10. Hematoxylin and Eosin (H&E) Staining for Follicle Count

The ovarian tissues underwent paraffin embedding, were fixed using 10% neutral buffered formalin (BBC, Washington, DC, USA), and were sectioned serially into slices that were 4 µm thick. Xylene and ethanol were used to deparaffinize the tissues of sectioned ovaries in a dry oven at 60 °C. Tissues that had been deparaffinized were washed in the sink. Following a 7 min immersion in Harris hematoxylin (Leica Biosystems, Wetzlar, Germany), the slides were counterstained with alcoholic eosin Y solution (Sigma-Aldrich). Whole ovarian stained slides were scanned using 3D HISTECH (The Digital Pathology Company, Budapest, Hungary). According to earlier reports, the total numbers of follicles, comprising primordial, primary, secondary, and pre-ovulatory follicles, as well as the antral follicles, were counted in serial sectioned slides at intervals of 100 μm.

### 4.11. Immunoflurescence Staining

Sections measuring 7 μm in thickness were cut from frozen ovary blocks and fixed in methanol for 10 min. Next, 1× phosphate-buffered saline (PBS) was used to wash the fixed tissues. The tissue edge was removed and placed in a humidified chamber. The blocking solution (Dako, Carpinteria, CA, USA) was applied to the tissues for 1 h at room temperature, and the primary antibody was applied to each tissue overnight in a 4 °C cold room. The following antibodies were mixed with antibody diluent buffer (Dako, Carpinteria, CA, USA) and used: rabbit anti-LC3b (2775S, Cell Signaling Technology) diluted 1:200 and rabbit anti-IGFBP2 (BS-1108R, Bioss) diluted 1:250. Next, all tissues were incubated at room temperature for 1 h. After three rounds of washing with 1× PBS at room temperature for 5 min each, the tissues were exposed for 1 h at room temperature to a secondary antibody. The tissues were washed three times for 5 min in 1× PBS at room temperature. Afterward, mounting medium with DAPI (VECTASHIELD, Burlingame, CA, USA) was used to mount the tissues. The stained slides were observed using fluorescence microscopy (Zeiss LSM 780, Oberkochen, Germany) at 40× magnification. Each slide was examined, and a representative image was selected. ImageJ (ImageJ 1.8 accessed on 22 May 2023) was used to analyze the tissues and cells.

### 4.12. Immunohistochemistry Staining

Xylene and ethanol were used to deparaffinize the tissues of sectioned ovaries in a dry oven at 60 °C. The deparaffinized tissues underwent antigen retrieval using an EDTA reaction (eLbio, Seongnam-Si, Republic of Korea), blocking in 3% hydrogen peroxide at room temperature, and primary antibody reaction overnight at 4 °C. The mouse anti-GLUT4 antibody (ab216661, Abcam) was diluted 1:250. The tissues on slides were incubated with Dako Real EnVision HRP Rabbit/Mouse secondary antibody (Dako, Carpinteria, CA, USA) at room temperature for 1 h after the unbound primary antibody was removed. The slides were counterstained with hematoxylin (Dako, Carpinteria, California, USA) after treatment with DAB (Dako, Carpinteria, CA, USA). The slides were washed with tap water after the reaction. Dehydration of the slides was accomplished using ethanol and xylene. The 3DHISTECH application (The Digital Pathology Company) was used to quantify the tissues.

### 4.13. Enzyme-Linked Immunosorbent Assay

The aortas of the rats in the normal, nontransplantation (NTx), and PD-MSC transplantation (Tx) groups were used to obtain all blood samples. A vacuum-sealed bag (BD Biosciences, San Jose, CA, USA) was used to collect blood, and from that blood batch serum samples were separated. Each rat serum sample was subjected to an ELISA kit analysis to determine the levels of ATP production (Thermo Fisher, Waltham, MD, USA), estrogen (Bio Vision, Milpitas, CA, USA), anti-Mullerian hormone (AMH; Elabscience Biotechnology, HX, USA), follicle-stimulating hormone (FSH; Abnova, Taipei, Taiwan), and testosterone (TES; CusaBio, Houston, TX, USA). All blood serum samples were stored at a temperature of −80 °C. The plates coated with antibodies contained the same volume of sample. Following the addition of the proper horseradish peroxidase (HRP) conjugates to each well, the wells were incubated at 37 °C. After the substrates were added and given time to develop in the dark, the antibody activity was measured using a microplate reader (BioTek, Winooski, VT, USA). 

### 4.14. The Statistical Analysis

Each experiment was carried out in two or three copies. The mean and standard error are the results that are displayed. Turkey’s post hoc test was used after Student’s *t*-test and a one-way ANOVA were used to evaluate the groupwise comparisons. PRISM 5.01 (GraphPad Software version 5.01, San Diego, CA, USA) was used to evaluate the data, and a *p* value of 0.05 indicated statistical significance.

## 5. Conclusions

Increased IGFBP2 by PD-MSCs triggers glucose metabolism via the AMPK signaling pathway and activates mitochondrial function in the ovarian follicles of the TAA-injured rat ovary model. Additionally, increased IGFBP2 cytokine production by PD-MSCs affects granulosa cells in ovarian follicles, meaning glucose imbalance and insulin resistance are ameliorated in granulosa cells. Finally, activated AMPK signaling improves not only glucose homeostasis but also ovarian function through decreased insulin resistance, although our study demonstrated that the IGFBP2–AMPK signaling pathway extends glucose uptake and follicular development. These findings may also have implications for new therapeutic stem cell strategies for infertile women with metabolic disorders.

## Figures and Tables

**Figure 1 ijms-24-16531-f001:**
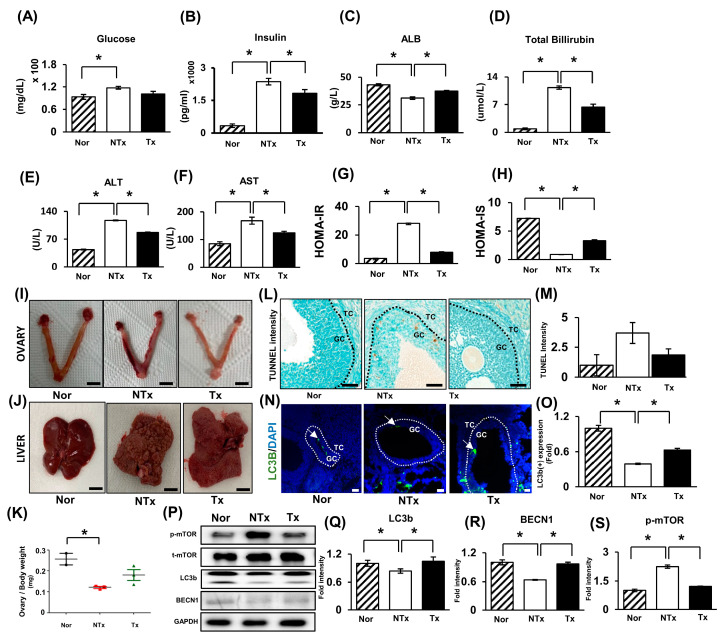
Thioacetamide-induced ovarian dysfunction and insulin resistance and impaired glucose metabolism. (**A**) Glucose, (**B**) insulin, (**C**) albumin, (**D**) total bilirubin, (**E**) ALT, and (**F**) AST levels were analyzed in serum by ELISA. (**G**,**H**) The HOMA-IR was performed according to the following equation: glucose (mg/dL) × insulin (mU/mL)/405; HOMA-IS was calculated according to the following equation: 1/HOMA-IR. (**I**) A morphological analysis in ovarian tissue of TAA- injury rats after sacrifice. (**J**) A morphological analysis in liver tissue of TAA-injury rats after sacrifice. Scale bar: 5 mm. (**K**) The ratio of ovarian weight to body weight analyzed after sacrifice. (**L**) The apoptotic cells of follicles were stained with a TUNEL assay kit. Scale bar: 100 μm; magnification: 20X. (**M**) The TUNEL intensity was quantified by 3D Histech. (**N**) The localization of LC3b protein expression in ovarian follicles analyzed using immunofluorescence (white arrow). Scale bar: 50 μm; magnification: 20X. (**O**) The localization of LC3b was quantified using ImageJ. (**P**) The protein expression was analyzed using western blotting.And its densitometry quantification showed the upregulation of (**Q**) LC3b and (**R**) BECN1 and the downregulation of (**S**) mTOR after PD-MSC transplantation. GC: granulosa cell; TC: theca cell. The data represent the mean ± S.D. Statistical significance was determined using a one-way ANOVA. Note: * *p* < 0.05, Nor vs. NTx; * *p* < 0.05, NTx vs. Tx.

**Figure 2 ijms-24-16531-f002:**
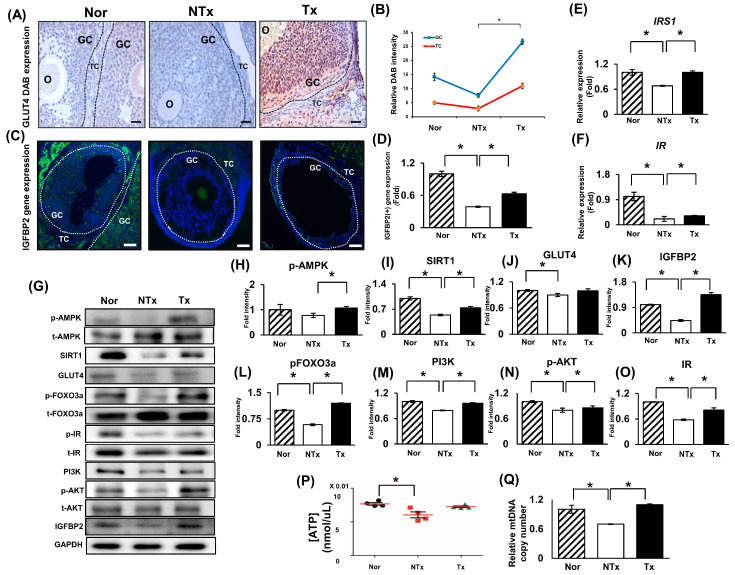
Glucose metabolism and insulin pathway signaling activated by PD-MSCs through the AMPK signaling pathway. Glucose metabolism and insulin pathway signaling are activated by PD-MSCs through the AMPK signaling pathway. (**A**) The localization of GLUT4 in mature follicles (e.g., antral follicles) of ovarian tissues was analyzed via immunohistochemistry staining. Scale bar: 100 μm; magnification: 40X. (**B**) The relative DAB intensity was analyzed by 3D Histech. (**C**) The localization of IGFBP2 in ovarian tissue was analyzed using immunofluorescence. Scale bar: 50 μm; magnification: 20X. (**D**) IGFBP2 gene expression was analyzed using the ImageJ program. The mRNA expression of IRS1 (**E**) and IR (**F**) was analyzed using qRT-PCR. Western blotting (**G**) and protein expression of AMPK (**H**), SIRT1 (**I**), GLUT4 (**J**), IGFBP2 (**K**), FOXO3a (**L**), PI3K (**M**), AKT (**N**), and IR (**O**) were quantified. (**P**) The concentration of ATP production was determined in serum via ELISA. (**Q**) The ratio of mitochondrial DNA in the gDNA of ovarian tissues was analyzed via qRT-PCR. GC: granulosa cell; TC: theca cell. The data represent the mean ± S.D. Statistical significance was determined using a one-way ANOVA. Note: * *p* < 0.05, Nor vs. NTx; * *p* < 0.05, NTx vs. Tx.

**Figure 3 ijms-24-16531-f003:**
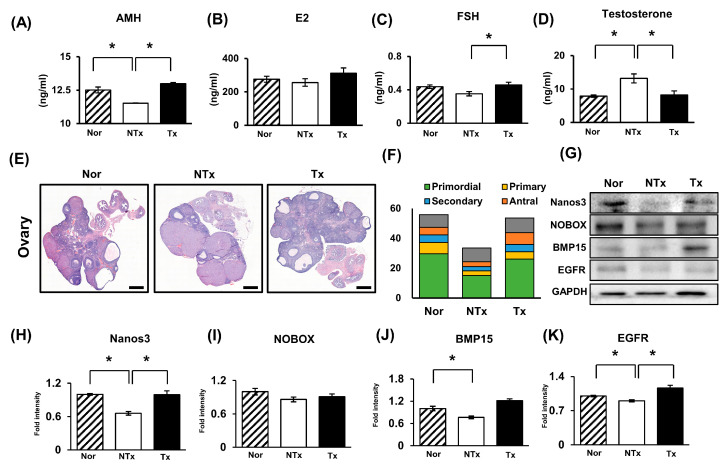
Effect of PD-MSCs on ovarian function in the ovaries of the TAA-injured rats. (**A**) The levels of AMH, (**B**) E2, (**C**) FSH, and (**D**) testosterone in individual serum samples were analyzed via ELISA. (**E**) A histological analysis of follicular development was performed using H&E staining. Scale bar: 2 mm; magnification: 1.4×. (**F**) The follicular count according to follicular development was analyzed via serial sectioned ovarian tissues. (**G**–**K**) The protein expression of Nanos3, Nobox, BMP15, and EGFR related to folliculogenesis was analyzed using Western blotting. The data represent the mean ± S.D. Statistical significance was determined using a one-way ANOVA. Note: * *p* < 0.05, Nor vs. NTx; * *p* < 0.05, NTx vs. Tx.

**Figure 4 ijms-24-16531-f004:**
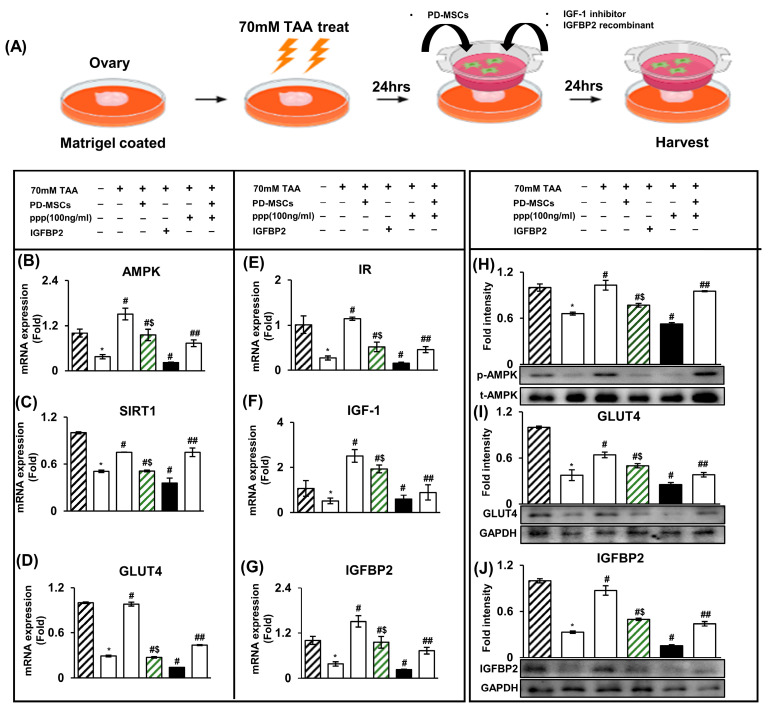
An increase in IGFBP2 cytokines secreted by PD-MSCs activated glucose metabolism in the TAA-injured rat ovary model. (**A**) Scheme of the ex vivo experiments. Ovaries were harvested and treated with PD-MSCs, recombinant IGFBP2, and PPP (IGF-1 inhibitor). (**B**–**G**) The mRNA expression of AMPK, SIRT1, GLUT4, IR, IGF-1, and IGFBP2 was analyzed using qRT-PCR. (**H**–**J**) The protein expression of AMPK, GLUT4, and IGFBP2 was analyzed via Western blotting. The data represent the mean ± S.D. Statistical significance was determined by using a one-way ANOVA. Note: * *p* < 0.05, Nor vs. TAA treatment; # *p* < 0.05, NTx vs. cocultivation with PD-MSCs; recombinant IGFBP2, inhibitor, $ *p* < 0.05 cocultivation with PD-MSCs vs. recombinant IGFBP2; ## *p* < 0.05 inhibitor vs. inhibitor + PD-MSCs.

**Figure 5 ijms-24-16531-f005:**
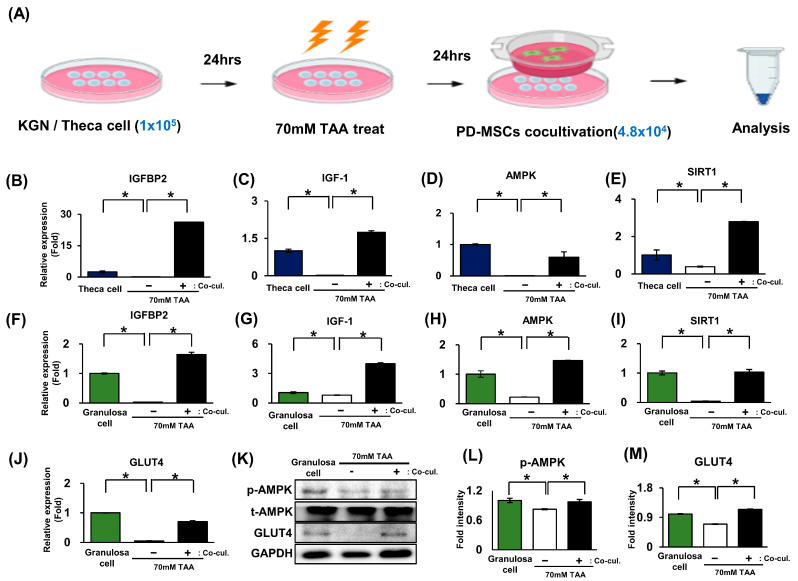
The activation of glucose metabolism markers was enhanced by the AMPK signaling pathway in granulosa cells (in vitro). (**A**) Scheme of the in vitro experiment. KGN cells and primary theca cells were seeded in 6-well plates and treated with 70 mM TAA and PD-MSCs. (**B**–**E**) In theca cells, the mRNA expression of IGFBP2, IGF-1, AMPK, SIRT1, and GLUT4 was analyzed via qRT-PCR. (**F**–**J**) In granulosa cells, the mRNA expression of IGFBP2, IGF-1, AMPK, and SIRT1 was analyzed via qRT-PCR. (**K**) The protein expression of AMPK and GLUT4 was analyzed using Western blotting. (**L**,**M**) The Western blot bands of AMPK and GLUT4 gene were quantified using ImageJ. The data represent the mean ± S.D. Statistical significance was determined by using a one-way ANOVA. Note: * *p* < 0.05, theca cell/granulosa cell vs. NTx; * *p* < 0.05, NTx vs. Tx.

**Table 1 ijms-24-16531-t001:** Primer sequences for quantitative real-time PCR.

Gene	Primer	Annealing Temperature (°C)	NM Number
*hIGFBP2*	F: 5′-ACA TCC CCA ACT GTG ACA AG-3′	56.8	NM_001313993.2
	R: 5′-ATC AGC TTC CCG GTG TTG A-3′		
*hAMPK*	F: 5′- CCA TGA AGA GGG CCA CAA TC-3′	58	NM_001355035.2
	R: 5′-TGC CAA AGG ATC CTG GTG AT -3′		
*hGLUT4*	F: 5’-ATC CTT GGA CGA TTC CTC ATT GG-3’	59.4	NM_175697.3
	R: 5’-CAG GTG AGT GGG AGC AAT CT -3’		
*hGAPDH*	F: 5’-CTC CTC TTC GGC AGC ACA-3’	58.2	NM_001357943.2
	R: 5’-AAC GCT TCA CCT AAT TTG CGT-3’		
*rIGFBP2*	F: 5’-GCA AAG GTG CCA AAC ACC TC-3’	59.3	NM_001310659.1
	R: 5’-TTC CAG AGG ACC CCG ATC AT -3’		
*rGLUT4*	F: 5’-TCA TCT TCA CCT TCC TAA-3’	49.1	NM_001039163.2
	R: 5’-CCT CAG TCA TTC TCA TCT -3’		
*rAMPK*	F: 5’-AGA TTG CAA AGG GCA TGA ACT AC-3’	57.8	NM_023991.1
	R: 5’-ACA TTC cTG CT GCC AAG TC-3’		
*rIR*	F: 5’-TTT TTG TCC CCA GGC CAT CC -3’	59.6	NM_017071.2
	R: 5’-CCT GTG CTC CTC CTG ACT TG-3’		
*rGAPDH*	F: 5’-TCT CTG CTC CTC CCT GTT CTA-3’	59.4	NM_017008.4
	R: 5’-ATG AAG GGG TCG TTG ATG GC -3’		

## Data Availability

All data and Supplementary Materials supporting the findings of this analysis are available from the corresponding author upon reasonable request.

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
