# Peer review of "Increased IGFBP2 Levels by Placenta-Derived Mesenchymal Stem Cells Enhance Glucose Metabolism in a TAA-Injured Rat Model via AMPK Signaling Pathway"

_ijms, 2023, doi:10.3390/ijms242216531_

Round 1
Reviewer 1 Report (New Reviewer)
Comments and Suggestions for Authors
The article established a rat injury model using thioacetamide (TAA). PD-MSCs were transplanted into the tail vein to explore the differences in glucose metabolism factors, follicular development indicators, and expression of sex hormones compared to the non-transplanted group. In addition, an in vitro co-culture group and a recombinant protein treatment group were also set up, and the comparison showed a significant increase in AMPK, glut4 mRNA, and protein expression. In conclusion, the elevated levels of IGFBP2 in PD-MSCs play a crucial role in glucose metabolism and ovarian function through the AMPK signaling pathway. However, there are several minor errors that are unacceptable, and some of the original WB images are unusable, requiring additional experiments. The following are some of the issues identified in the article:
1. Lines 60: The first line should be indented by two spaces.
Lines 65-67: Only half of the parentheses appeared. Where does the content in parentheses end?
2. Lines 73-77: Can the section introducing PCOS be rearranged to the beginning? It is suggested to reorganize the logical flow of the introduction for better coherence.
3. More information about the applications of PD-MSCs needs to be added; its introduction appears somewhat abrupt. What is the HOMA-IS index, and how is it calculated?
4. Are the Section 2.4 title and Section 2.5 title the same?
5. The images are relatively small; it is recommended to enlarge them.
6. Introduction of BECN1 gene and KGN cells when they first appear?
7. The discussion section provides relatively little discussion of the study's results; it is advisable to increase the discussion."
8. The original WB images of the BECN1 gene in Figure 1 and the BMP15 gene in Figure 3 were not usable; additional experiments should be conducted to support the conclusion. Otherwise, the results are not valid.
Author Response
Reviewer #1:
Comments and suggestions for authors:
The article established a rat injury model using thioacetamide (TAA). PD-MSCs were transplanted into the tail vein to explore the differences in glucose metabolism factors, follicular development indicators, and expression of sex hormones compared to the non-transplanted group. In addition, an in vitro co-culture group and a recombinant protein treatment group were also set up, and the comparison showed a significant increase in AMPK, glut4 mRNA, and protein expression. In conclusion, the elevated levels of IGFBP2 in PD-MSCs play a crucial role in glucose metabolism and ovarian function through the AMPK signaling pathway. However, there are several minor errors that are unacceptable, and some of the original WB images are unusable, requiring additional experiments. The following are some of the issues identified in the article:
Author’s reply:
We greatly appreciate the reviewer’s critical statement that “Increased IGFBP2 levels by placenta-derived mesenchymal stem cells enhance glucose metabolism in a TAA-injured rat model via AMPK signaling pathway”.
Point #1: Lines 60: The first line should be indented by two spaces.Lines 65-67: Only half of the parentheses appeared. Where does the content in parentheses end?
- Author’s response:
Thank you for your critical comments. As your comment, I modified the manuscript.
Point #2: Lines 73-77: Can the section introducing PCOS be arranged to the beginning? It is suggested to reorganize the logical flow of the introduction for better coherence.
- Author’s response:
Thank you for your critical comments. Based on your comments, we have reorganized the section introducing PCOS to make it easier for readers to understand the logical flow of the introduction.
Point #3: More information about the applications of PD-MSCs needs to be added: its introduction appears somewhat abrupt. What is the HOMA-IS index, and how is it calculated?
- Author’s response:
We greatly appreciate the reviewer bringing up this important point. Several types of MSCs (BM-MSCs, UC-MSCs etc.) also improve ovarian function. To logically explain why we used PD-MSCs, we added a mention of the benefits of PD-MSCs and their efficacy in in vivo experiments.
Also, HOMA-IS (homeostatic model assessment-insulin sensitivity) is inversely proportional to HOMA-IR, and commonly known as an index for evaluating insulin sensitivity in clinical practice.HOMA-IS is calculated as “1/HOMA-IR”, and HOMA-IR is calculated as “(fasting glucose x fasting insulin)/405” [Albareda el al. Diabetologia 2000.].
- Albareda M; Rodrigue-Espinosa J; Murugo M; Leiva A, Corcoy R. Demeestere. Assessment of insulin sensitivity and beta-cell function from measurements in the fasting state and during an oral glucose tolerance test. Diabetologia.2000;43(12):1507-11.
Point #4: Are the Section 2.4 title and Section 2.5 title the same?
- Author’s response:
We appreciate the reviewer for pointing out an important point. First of all, we apologize for any confusion.
Section 2.4 presents the results of analysis using IGFBP2 recombinant and IGF-1 inhibitor to analyze the paracrine effect on IGFBP2 secretion by PD-MSCs in ex-vivo experiments. Therefore, the 2.4 title was modified to “The increase in IGFBP2 secreted by PD-MSCs triggers glucose metabolism in TAA treated-explanted ovaries with IGF-1 inhibitor or IGFBP2 recombinant (ex-vivo)”.
Section 2.5 shows that AMPK signaling is activated through IGFBP2 secreted by PD-MSCs in TAA-treated KGN cells , improving glucose metabolism. Therefore, the 2.5 title was modified to “The activation of glucose metabolism markers was enhanced by the AMPK signaling pathway in granulosa cells (In-vitro)”.
Point #5: The images are relatively small: it is recommended to enlarge them.
- Author’s response:
Thank you for your critical comments. Based your comment, we edited the image to make it look better.
Point #6: Introduction of BECN1 gene and KGN cells when they first appear?
- Author’s response:
Thank you for your critical comments. Based your comment, we added the content about BECN1 and KGN cell in the paper (result part)
Point #7: The discussion section provides relatively little discussion of the study`s results; it is advisable to increase the discussion.
- Author’s response:
Thank you for your critical comments. Thank you for your critical comments. As your comment, we discuss about our paper more deeply.
Point #8: The original WB images of the BECN1 gene in Figure 1 and the BMP15 gene in Figure 3 were not usable; additional experiments should be conducted to support the conclusion. Otherwise, the results are not valid.
- Author’s response:
Thank you for your critical comments. We replaced the WB image with a clear band through additional experiments.
Figure 1)
Left figure: origin figure
Right figure: new figure to be replaced
Original image
Figure 3)
Left figure: origin figure; Right figure: new figure to be replaced
Original image

Reviewer 2 Report (New Reviewer)
Comments and Suggestions for Authors
The main question addressed by the research is how increased IGFBP2 levels can enhance glucose metabolism via the AMPK signaling pathway. Research on glucose metabolism in the ovaries is important for understanding important processes such as reproductive functions, follicle development and ovulation. However, it is noteworthy that the number of studies on this subject is limited. In this context, the study presented will provide significant knowledge in this field, where there are a limited number of studies.
The research findings on the effects of placenta-derived mesenchymal stem cells on glucose metabolism in a rat model can be considered relevant in the field of reproductive medicine and metabolic disorders. Although it is accepted that the findings obtained from the study will have an impact that may contribute to potential treatment strategies for infertile women with metabolic disorders, the main purpose of this study is to investigate the effects of placenta-derived mesenchymal stem cells on glucose metabolism in the rat model and the underlying mechanisms in this process.
The article presents original research findings on the effects of placenta-derived mesenchymal stem cells on glucose metabolism in a rat model. The study contributes to the existing literature by providing new insights into the underlying mechanisms involved in this process, specifically the role of increased IGFBP2 levels and the AMPK signaling pathway. Therefore, this article adds to the subject area by providing new knowledge and potential avenues for future research.
The study's methodology involves the use of a rat model and the collection of data on glucose metabolism, insulin resistance, and other related factors. The authors also conducted various assays to examine the underlying mechanisms involved in the process. The article provides a detailed description of the methods used, including the source of the stem cells, animal care, and statistical analysis.
However, although the use of a larger sample in the study and the inclusion of a negative control group would allow the findings to be tested with higher accuracy, the material method of the presented study was presented in sufficient detail.
The conclusions appear to address the main question posed in the study, which is to investigate the effects of placenta-derived mesenchymal stem cells on glucose metabolism and explore the underlying mechanisms involved in this process.
The literature used in the study is current and sufficient.
Author Response
Reviewer #2:
Comments and suggestions for authors:
The main question addressed by the research is how increased IGFBP2 levels can enhance glucose metabolism via the AMPK signaling pathway. Research on glucose metabolism in the ovaries is important for understanding important processes such as reproductive functions, follicle development and ovulation. However, it is noteworthy that the number of studies on this subject is limited. In this context, the study presented will provide significant knowledge in this field, where there are a limited number of studies.
The research findings on the effects of placenta-derived mesenchymal stem cells on glucose metabolism in a rat model can be considered relevant in the field of reproductive medicine and metabolic disorders. Although it is accepted that the findings obtained from the study will have an impact that may contribute to potential treatment strategies for infertile women with metabolic disorders, the main purpose of this study is to investigate the effects of placenta-derived mesenchymal stem cells on glucose metabolism in the rat model and the underlying mechanisms in this process.
The article presents original research findings on the effects of placenta-derived mesenchymal stem cells on glucose metabolism in a rat model. The study contributes to the existing literature by providing new insights into the underlying mechanisms involved in this process, specifically the role of increased IGFBP2 levels and the AMPK signaling pathway. Therefore, this article adds to the subject area by providing new knowledge and potential avenues for future research.
The study's methodology involves the use of a rat model and the collection of data on glucose metabolism, insulin resistance, and other related factors. The authors also conducted various assays to examine the underlying mechanisms involved in the process. The article provides a detailed description of the methods used, including the source of the stem cells, animal care, and statistical analysis.
However, although the use of a larger sample in the study and the inclusion of a negative control group would allow the findings to be tested with higher accuracy, the material method of the presented study was presented in sufficient detail. The conclusions appear to address the main question posed in the study, which is to investigate the effects of placenta-derived mesenchymal stem cells on glucose metabolism and explore the underlying mechanisms involved in this process. The literature used in the study is current and sufficient.
Author’s reply:
We greatly appreciate the reviewer’s positive statement that “Increased IGFBP2 levels by placenta-derived mesenchymal stem cells enhance glucose metabolism in a TAA-injured rat model via AMPK signaling pathway”.We studied the importance of glucose metabolism in the ovary through stem cell therapy based on the latest papers, although this is a limited research topic. Based on your comments, we will conduct further studies with larger samples and negative controls to analyze the mechanism more logically.

Reviewer 3 Report (New Reviewer)
Comments and Suggestions for Authors
This study interrogated the effect of PD-MSCs on glucose metabolism in a TAA-injured model and the correlation between glucose and ovarian function. Overall, this study was well-designed, and the conclusions are fully supported by the existing data. Please see below my minor concerns.
1. In the abstract, there is no description about how IGFBP2 was regulated, and suddenly it appeared in the conclusion.
2. Couldn't see the error bars in Fig 2B.
3. Better representative WB images are required in Fig 3G.
4. Better representative WB images are required in Fig 4H. Also, protein names are missing in panel H,I and J.
Comments on the Quality of English LanguageThe overall interpretation are clear and understandable.
Author Response
Reviewer #3:
Comments and suggestions for authors:
This study interrogated the effect of PD-MSCs on glucose metabolism in a TAA-injured model and the correlation between glucose and ovarian function. Overall, this study was well-designed, and the conclusions are fully supported by the existing data. Please see below my minor concerns.
Author’s reply:
Thank you for providing this insight about “Increased IGFBP2 levels by placenta-derived mesenchymal stem cells enhance glucose metabolism in a TAA-injured rat model via AMPK signaling pathway”.
Point #1: In the abstract, there is no description about how IGFBP2 was regulated, and suddenly it appeared in the conclusion.
- Author’s response:
Thank you for your critical comments. As your comments, we added function of IGFBP2 in abstract Additionaly we showed PD-MSCs secreted more IGFBP2 than other metabolic cytokine.
Point #2: Couldn't see the error bars in Fig 2B.
- Author’s response:
Thank you for your critical comments.We appreciate the reviewer for pointing out an important point. First, we apologize for any confusion. The issue occurred while adjusting the fugure to be small. As your comment, we revised the error bar in Fig 2B.
Left figure: origin figure; Right figure: new figure to be replaced
Point #3: Better representative WB images are required in Fig 3G.
- Author’s response:
Thank you for your critical comments.We have replaced more representative WB images performed additional experiment.
Left figure: origin figure; Right figure: new figure to be replaced
Original image
Point #4: Better representative WB images are required in Fig 4H. Also, protein names are missing in panel H,I and J.
- Author’s response:
Thank you for your critical comments. We have replaced more representative WB images performed additional experiment
and have reflected this comment by “protein names are missing in panel I,H and J”
Left figure: origin figure; Right figure: new figure to be replaced

Round 2
Reviewer 1 Report (New Reviewer)
Comments and Suggestions for Authors
All questions answered.
Author Response
IJMS-2698203MinorRevision
Increased IGFBP2 levels by placenta-derived mesenchymal stem cells enhance glucose metabolism in a TAA-injured rat model via AMPK signaling pathway
Academic editor #1:
"This study is interesting and provides novel findings. Moreover, the Authors have amended their work incorporating the technical requests and
suggestions raised by the Reviewers. Nevertheless, the text contains grammar errors, and needs frequent re-writing of sentences, particularly in the discussion section. As a result the manuscript cannot be published in its present form!
Here are just a few examples:
- Figure 2. The glucose metabolism and insulin pathway
sugnaling.....(page 6)
- 2.2. The glucose metabolism and insulin pathway sugnaling.....
- Page 2, lines 68-70, sentence needs to be rephrased.
- Page 3, line 123: To confrim,
- Page 3, line 142: Additionally, To confrim
- Page 12 line 416: To confrim
- Page 12 line 409: To confrim
- Page 12 line 338: As preliminary study, we found that more IGFBP2 secreted by PD-MSCs: needs rephrasing.
Page 12, lines 342-343: It is especially notable that improving ovarian function for moderating metabolic disorder via PD-MSCs. Needs rephrasing
Page 12, lines 368-369: Also, we found that anti-apototic effect and increasing autophagy in ovarian tissue through transplantation PD-MSCs. Needs rephrasing.
Page 12, lines 370-371: This result indicated that TAA induces metabolic dysfuncton and apototicm low autophagic status. Grammar errors
Page 12, line 372: Futher, this figure suggests that possibility of TAA inducing ovarian dysfunction. The phrasing is badly connected to the upper part of the text.
Page 12, line 374: As shown in Figure 2, we confirm that glucose metabolism related markers using by staining and molecular works. Could be better rephrased.
Page 12, lines 399-400. To confirm the effect on follicular development, we evaluted serially ovarian tissues from each group by H&E (figure 3). Then, each follicle stage was counted by three different people. Sentences like this should not appear in a Discussion section.
The whole discussion section is fragmented and needs extensive rewriting."
Author’s reply:
Thank you for your critical comments. As your comment, I modified the manuscript.
- Author’s response:
Thank you for your critical comments. First, we apologize for any confusion about grammer error and format.Based on your comments, we have reorganized the manuscript to make it easier for readers to understand.
Reviewer #1:
Comments and suggestions for authors:
All questions answered.
Author’s reply:
We greatly appreciate the reviewer’s critical statement that “Increased IGFBP2 levels by placenta-derived mesenchymal stem cells enhance glucose metabolism in a TAA-injured rat model via AMPK signaling pathway”.

This manuscript is a resubmission of an earlier submission. The following is a list of the peer review reports and author responses from that submission.
Round 1
Reviewer 1 Report
Comments and Suggestions for Authors
Through molecular investigations, this study examined the effects of metformin on ovarian function in rats. Additionally, it revealed the regulatory influence of PD-MSC transplantation on essential genes and proteins in ovarian tissue, providing valuable insights for further research on the mechanisms underlying ovarian function regulation. I recommend the manuscript could be accepted for publication after minor revision.
Comment 1: Please set the p-values in the entire text to italic.
Comment 2: Please align all the figures in the manuscript with their respective scales/rulers.
Comment 3: Please provide the gene expression level data that corresponds to the protein expression level.
Comment 4: Please explain the specific inconsistencies between the concentrations of IGF-1 inhibitor PPP and recombinant IGFBP2 in line 264 of the manuscript compared to those in line 472.I would appreciate a detailed explanation.
Comment 5: Intraperitoneal injections can induce stress in animals, and instead of habituation, they tend to show sensitization with repeated injections. Therefore, should a 12-week period of intraperitoneal injection be considered as part of the stress protocol? Moreover, how much potential impact could this intervention have on the study results?
Comment 6: Please check the references. The review must reflect the latest research on others, and if the quoted documents are obsolete references a few years ago, they can not reflect the latest research trends.
Comment 7: The following studies may serve as references for your research.
[1] Xiaoyu Wang, Chenghong Xing, Guyue Li, Xueyan Dai, Xiaona Gao, Yu Zhuang, Huabin Cao, Guoliang Hu, Xiaoquan Guo, Fan Yang. The key role of proteostasis at mitochondria-associated endoplasmic reticulum membrane in vanadium-induced nephrotoxicity using a proteomic strategy. Sci Total Environ, 2023. 869:161741. doi: 10.1016/j.scitotenv.2023.161741
[2] Shixuan Lin, Fan Yang, Mingwen Hu, Jing Chen, Guiping Chen, Aiming Hu, Xiong Li, Danghua Fu, Chenghong Xing, Zhiwei Xiong, Yunhui Wu, Huabin Cao. Selenium alleviates cadmium-induced mitophagy through FUNDC1-mediated mitochondrial quality control pathway in the lungs of sheep. Environ Pollut, 2023, 319:120954. doi: 10.1016/j.envpol.2022.120954.
[3] Xue-Yan Dai, Shi-Yong Zhu, Jian Chen, Mu-Zi Li, Yi Zhao, Milton Talukder and Jin-Long Li. Lycopene alleviates di(2-ethylhexyl) phthalate-induced splenic injury by activating P62-Keap1-NRF2 signaling. Food and Chemical Toxicology. 2022 Jul 31;168:113324.
[4]Yukun Fang, Chenghong Xing, Xiaoyu Wang, Huabin Cao, Caiying Zhang, Xiaoquan Guo, Yu Zhuang, RuiMing Hu, Guoliang Hu, Fan Yang.Activation of the ROS/HO-1/NQO1 signaling pathway contributes to the copper-induced oxidative stress and autophagy in duck renal tubular epithelial cells.Sci Total Environ. 2021 Feb 25;757:143753. doi: 10.1016/j.scitotenv.2020.143753.
Comments on the Quality of English LanguageTitle: Increased IGFBP2 levels by placenta-derived mesenchymal stem cells enhance glucose metabolism in a TAA-injured rat model via AMPK signaling pathway.
Through molecular investigations, this study examined the effects of metformin on ovarian function in rats. Additionally, it revealed the regulatory influence of PD-MSC transplantation on essential genes and proteins in ovarian tissue, providing valuable insights for further research on the mechanisms underlying ovarian function regulation. I recommend the manuscript could be accepted for publication after minor revision.
Comment 1: Please set the p-values in the entire text to italic.
Comment 2: Please align all the figures in the manuscript with their respective scales/rulers.
Comment 3: Please provide the gene expression level data that corresponds to the protein expression level.
Comment 4: Please explain the specific inconsistencies between the concentrations of IGF-1 inhibitor PPP and recombinant IGFBP2 in line 264 of the manuscript compared to those in line 472.I would appreciate a detailed explanation.
Comment 5: Intraperitoneal injections can induce stress in animals, and instead of habituation, they tend to show sensitization with repeated injections. Therefore, should a 12-week period of intraperitoneal injection be considered as part of the stress protocol? Moreover, how much potential impact could this intervention have on the study results?
Comment 6: Please check the references. The review must reflect the latest research on others, and if the quoted documents are obsolete references a few years ago, they can not reflect the latest research trends.